# Utilization of Microbial Consortia as Biofertilizers and Biopesticides for the Production of Feasible Agricultural Product

**DOI:** 10.3390/biology10111111

**Published:** 2021-10-28

**Authors:** Renganathan Seenivasagan, Olubukola Oluranti Babalola

**Affiliations:** Food Security and Safety Niche Area, Faculty of Natural and Agricultural Sciences, North-West University, Mmabatho 2735, South Africa; srinide@gmail.com

**Keywords:** bioinoculant, biopesticides, PGPR, microbial inoculants, organic farming, yield component

## Abstract

**Simple Summary:**

Recently in agriculture, the usage of chemical pesticides and fertilizers has increased tremendously. Additionally, it shows severe effects on human health, ecosystem, and groundwater. Environment-friendly methods are used to improve soil fertility, pests, and disease control. Biopesticide and biofertilizers have the future to upgrade sustainable agriculture for many years. This review highlights the efficacy of biofertilizers and biopesticides in improving crop yielding. It provides an eco-friendly and cost-effective method to get more yield for farmers. It describes the prominence of microbial inoculants in plant cultivation.

**Abstract:**

Farmers are now facing a reduction in agricultural crop yield, due to the infertility of soils and poor farming. The application of chemical fertilizers distresses soil fertility and also human health. Inappropriate use of chemical fertilizer leads to the rapid decline in production levels in most parts of the world, and hence requires the necessary standards of good cultivation practice. Biofertilizers and biopesticides have been used in recent years by farmers worldwide to preserve natural soil conditions. Biofertilizer, a replacement for chemical fertilizer, is cost-effective and prevents environmental contamination to the atmosphere, and is a source of renewable energy. In contrast to chemical fertilizers, biofertilizers are cost-effective and a source of renewable energy that preserves long-term soil fertility. The use of biofertilizers is, therefore, inevitable to increase the earth’s productivity. A low-input scheme is feasible to achieve farm sustainability through the use of biological and organic fertilizers. This study investigates the use of microbial inoculants as biofertilizers to increase crop production.

## 1. Introduction

Chemical fertilizers and pesticide dependence in conventional agriculture have increased, due to the significant growth of the human population and food demands [1]. Plant nutrition plays a major role in the increased demand for food supply. An increase in crop production has made it possible through the use of commercial artificial fertilizers. Phosphorus, nitrogen, and potassium fertilizer have frequently increased for crop production and agricultural systems with low cost [2]. Soil quality deterioration reversed biodiversity, and increased water and air pollution, and human health has also created excess use of chemical fertilizer [3]. The agriculture ecosystem, soil fertility, and cultivated crop growth get affected, due to excessive usage of chemical pesticides [4]. To overcome such drawbacks, a biofertilizer, a biological agent, is used for convalescing this problem. The loss of topsoil, soil infertility, plant growth reduction, reduced yield index, and gradual decrease of indigenous microbial diversity could be managed by microbial inoculants using agricultural practice. Pesticides and chemical fertilizers create environmental issues that microbial inoculants can comfortably overcome, which serves as a potential alternative and psychostimulants [5,6].

For a healthy environment, the management of integrated nutrient systems and sustained agricultural productivity is greatly influenced by microbial inoculants [7]. Microbial inoculants or biofertilizers contain living microorganisms that colonize the rhizosphere and helps in the promotion of plant growth. The converts the insoluble elements in the soil to a soluble form by a biological process similar to rock phosphate solubilization and nitrogen fixation [8]. Beneficial microorganisms utilized in biofertilizers improve microflora, soil health, plant growth, plant disease control, and protect the plant from pests [9]. There are beneficial microbial inoculants, such as nitrogen fixer, phosphate, sulfur, zinc solubilizer (VAM), and plant growth-promoting rhizobacteria, in biofertilizers. Plant growth-promoting rhizobacteria are bacteria that live freely on rhizosphere soil and promote plant growth. They also acts as biopesticides, based upon the ability or behavior of the crops and biocontrol agents (Nitrogen fixer, PSB, and SSB) [10].

‘Biopesticide’ implies the use of beneficial microorganisms to control the insects. However, the major constraint is the availability of biopesticides relative to the total cropped area. Specific pesticides, derived from natural materials, act as biopesticides, such as plants, animals, bacteria, and certain minerals [11]. Biopesticides are classified into 3 major categories: biochemical, plant, and microbial pesticides. All over the world, there are 90% of all biopesticides utilized. The most commercially successful biopesticide in the market is *Bacillus thuringiensis* (Bt) [12]. Modern agriculture requires biopesticide and biofertilizers, due to the demand for safe and residue-free crop production [3]. Therefore, to cater to the need, it is necessary that government, and nongovernment organizations should promote entrepreneurs for biofertilizers and biopesticides production.

The objective of this review is the improvement of plant growth and yield through various microbes, such as bacterial, fungal, virus, and algae inoculants as biofertilizer (nitrogen fixers, phosphate solubilizer (PSB), sulfur solubilizer, PGPR, VAM, and Azolla), PGPR (phosphate solubilizer, N_2_ fixers, phytohormones, siderophores, and antibiotics) and biopesticides (microbial, plant incorporated protection, and biochemical).

## 2. Biofertilizer

Microbial inoculants or biofertilizers are preparation containing viable algae, fungi, and bacteria alone or consortium together to support the plant growth and increase crop yield [13]. Biofertilizers contain beneficial microbes that improve soil chemical and biological characteristics by fixing nitrogen, cellulolytic activity, or phosphate. When they are applied to seed, plant surfaces, root, or soil, they inhabit the rhizosphere, and through their biological activity, they enhance nutrient bioavailability, promote plant’s growth, and increase the soil microflora. Thereby, they are preparations that readily improve the fertility of the soil [14,15]. Rhizobium has symbiotic associations with legume roots, such as rhizobacteria, that reside on the surface of the root or in the soil of the rhizosphere. Broad-spectrum biofertilizers include Blue-Green Algae (BGA), Rhizobium, and Azolla are crop-specific bio inoculants, such as *Azospirillum*, *Azotobacter*, phosphorus solubilizing bacteria (PSB), vesicular-arbuscular mycorrhiza (VAM), and Anabaena, as nitrogen-fixing cyanobacteria [15,16]. These bacteria are known as biofertilizers and plant growth-promoting rhizobacteria (PGPR). Competition mechanisms and antagonism activity are carried out by the enzymatic activity of PGPR for crop production, such as the inhibition of phytohormones and phytoparasites; it also helps plants in withstanding stress by heavy metal contaminations and pollutants [17,18].

Biofertilizers are eco-friendly, cost-effective, and can be produced in bulk on the farm itself if necessary. The crop yield is increased by 10–40% and up to 50 percent of nitrogen is fixed. The continuous application of biofertilizer of the land for 3–4 years can retain fertility, due to the efficiency of parental inoculums, which could maintain the growth and multiplication of plants effectively. They improve soil texture, pH, and other properties [19].

Biofertilizers are renewable sources of plant nutrients complementing chemical fertilizers at a low cost. In comparison to chemical fertilizers, biofertilizers are environmentally friendly; can be produced from natural sources, prevented from damage but also helps in building up healthy soil, and to some extent, plants are cleansed from chemical fertilizers that are precipitated [20]. Considering biofertilizer as a modern tool for agriculture, its use is vital as components of integrated nutrient management, reduction in the usage of hazardous chemicals, cost-effective, and source of renewable energy for plants in sustainable agriculture [21] (Figure 1 and Table 1).

### 2.1. Nitrogen Fixers

Nitrogen fixation is a dynamic and high-energy demanding process [26]. Elemental nitrogen conversion by biological nitrogen fixation is one way of converting into a plant’s usable form. Organic compounds are transformed into inert atmospheric N_2_ by nitrogen-fixing bacteria [27]. In biofertilizers, nitrogen fixers or N_2_ fixing species are used as fertilizers containing living microbial inoculants or microorganism classes. Microbial inoculants, such as Azotobacter, Rhizobium, Blue-Green Algae (BGA), Azospirillum, and Azolla, are used as biofertilizers, which help in nitrogen fixation by converting atmospheric nitrogen to plant useable form. Legume plants have root nodules inhabiting bacteria belonging to the genera *Sinorhizobium, Azorhizobium, Rhizobium, Bradyrhizobium*, and *Mesorhizobium*, collectively called rhizobia [22]. When rhizobial culture is inoculated in the field, rhizobial symbiosis occurs, increasing the yield of pulse crops up to 15–20 kg N/ha by rhizobium, and crop yield is increased by up to 20% [28].

By nature, Azotobacter has a major role in the nitrogen cycle as it has a range of metabolic capabilities [29]. Along with nitrogen-fixing, Azotobacter also produces vitamins, such as riboflavin and thiamine [30], indole acetic acid (IAA), cytokinins (CK), and Gibberellins (GA), via plant hormones [31]. Atmospheric nitrogen is fixed and supplied as ammonium by *Azotobacter chroococcum*. Therefore changing over of ammonium ions improves plant development by root architecture advancing and seed germination enhancement [32]. Azotobacter is also used to kill pathogenic microorganisms surrounding crop plant root systems [33]. Azospirillum is another aerobic, free-living, motile, gram-negative bacterium that can thrive under flooding conditions [34], supporting various aspects of plant growth and development [35]. Infield trials and greenhouse experiments with Azospirillum species, such as Azospirillum, including *A. irakense, A. lipoferum, A. halopraeferens, A. amazonense,* and *A. brasilense*, shows improved crop yield and plant growth [34] (Table 2). Plants inoculated with Azospirillum showed higher water and mineral uptake leads to better yield [22]. Hungria et al. [36] reported that *Azospirillum brasilense* is competent enough to promote the growth of plants by fixation of nitrogen, which helps to save money.

For the global nitrogen cycle, cyanobacteria are very necessary for significant N_2_ fixers on earth [46]. Cyanobacteria, mostly used as nitrogen-fixing biofertilizers, includes Scytonema, Tolypothrix, Plectonema, Aulosira, Anabaena, and Nostoc [47,48]. Along with releasing growth-promoting substances, *Cylindrospermum musicola* also releases vitamins and nitrogen. In rice plants, it also improves root growth and yield [49]. Crop plants inoculated with Rhizobium sp. showed a substantial increase in growth and yield, through a high number of root nodules, compared to uninoculated plants [50].

### 2.2. Phosphate Solubilizing Microorganisms

Plant growth and metabolism processes are mainly affected by the nutrient nitrogen followed by phosphate [51]. In virtually all major metabolic processes, such as respiration, photosynthesis, energy accumulation, and transfer, signal transduction, cell enlargement, cell division, and macromolecular biosynthesis, they play an important role. Phosphate contributes to the resistance to disease and helps to survive winter rigors in plants [52,53]. As it is present in the form of insoluble phosphates, approximately 95–99 percent of soil phosphorous is unusable for plants [54]. The P-solubilizing potential of microbial inoculants (biofertilizers) is used as an environmentally safe alternative to further chemical-based P fertilizer applications in agricultural soil [55]. Phosphorous can be solubilized by many microorganisms, including bacteria, fungi, actinomycetes, and even algae, such as Cyanobacteria and Mycorrhiza [1,56].

The most popular inoculants for phosphorus solubilizing bacteria (PSB) belong to the genera *Pseudomonas* spp. and Bacillus [57,58]. Other bacteria identified include *Serratia*, *Rhodococcus*, *Chryseobacterium*, *Phyllobacterium*, *Arthrobacter*, *Delftia* sp., *Gordonia*, [37], *Xanthomonas* [38], *Azotobacter* [39], *Enterobacter*, *Pantoea*, *Klebsiella* [40,41], Vibrio *proteolyticus* [42], *Beijerinckia*, *Burkholderia*, *Erwinia*, *Flavobacterium*, *Microbacterium* and *Rhizobium* [35], *Xanthobacter agilis* [43]. By releasing complexing or mineral dissolving compounds, such as (i) organic acid anions, protons, siderophores, CO_2_, and hydroxyl ions; (ii) extracellular enzyme release; and (iii) substrate degradation and P release, the soil microorganism employs P-solubilization mechanisms [59]. Organic acids are low molecular weight, such as citric and gluconic acids, which are synthesized by PSB during inorganic P solubilization [60]. The phosphate with the chelating cations of carboxyl and hydroxyl groups binds with organic acids, thereby releasing soluble phosphate and inducing soil acidification [61]. Heavy metal immobilization is performed by phosphate fertilizers. Microorganisms and plants solubilize insoluble phosphate compounds using their phosphatase enzyme and organic acids [62,63].

Phosphate solubilizing and stress-tolerant bacteria *Burkholderia vietnamiensis* produces gluconic acids, and 2-ketogluconic, which is involved in solubilizing phosphate [62]. Tomar et al. [64] reported that black gram (*Vigna mungo*) and lentil (*Lens esculentus*) inoculated with *B. firmus*, a phosphate solubilizing bacteria showed significant results in the increase of seed yield. Chabot et al. [65] described that the inoculation of P-solubilizing *Rhizobium leguminosarum* leads to the growth of maize and lettuce. Some fungi, such as Penicillium and Aspergillus, act as phosphorus solubilizers [53]. Mittal et al. [66] isolated six P-solubilizing fungi, four strains of *Penicillium citrinum*, and two strains of *A. awamori*, from the rhizosphere of various crops. *A. awamori* showed shoot height increase of 7–12%, seed number increase to three-fold, and increase in seed weight to two-fold, in comparison to uninoculated plants. Hajra et al. [67] reported that mycorrhizal plants had increased plant height and leaf area, in comparison to non-mycorrhizal plants, also showed a sharp decrease of nematode infection in plants (Table 2).

### 2.3. Potassium Solubilizing Microorganisms

Potassium is the third important plant growth nutrient that plays a vital role in plant metabolism, growth, and development. The plants would have poorly formed roots, grow slowly, produce small seeds and have lower yields without a sufficient supply of potassium [68] and increased vulnerability to diseases [69] and pests [70]. Potassium solubilizing microbes produce organic acids that can solubilize potassium rock [71]. Rhizosphere soil microbial inoculants, including *Aspergillus, Bacillus* sp., *Clostridium, Burkholderia, Acidothiobacillus ferrooxidans, Pseudomonas, Paenibacillus* sp., *Bacillus mucilaginosus*, *B. circulans*, and *B. edaphicus* has been reported to be released from potassium-bearing minerals in soils in an accessible form [72].

Organic acids are produced and secreted by microbial inoculants, such as *Bacillus mucilagenosus* and *Bacillus edaphicus*, in the solubilization of rock potassium [73]. Potassium solubilizing bacteria (KSB) beneficial effects have been reported on the growth of grape and cotton [74], sorghum [75], wheat [76], sudangrass [77], cucumber and pepper [66]. The significant mobilization of high potassium from waste mica, which acted as a potassium source for plant growth, resulted in wheat plants with *Bacillus mucilaginosus*, *Azotobacter chroococcum*, and *Rhizobium* [78].

### 2.4. Sulfur Dissolving Microorganisms

For the growth and development of plants, sulfur is one of the sixteen elements and the fourth main nutrient in crop production, after N, P, and K. As a result of microbial activity, which includes mineralization, immobilization, oxidation, and reduction processes, sulfur transformations in the soil. Sulfur oxidizing bacteria synthesis of organic compounds from carbon dioxide by sulfur oxidation process and produce sulfuric acids. Enzyme sulfatase was used in the catalyzation of sulfur compound mineralization and transformation into forms accessible to plants [79]. After the inoculation of sulfur-oxidizing bacteria (*Thiobacillus*), seeds of high S-demanding crops have proved to be very effective, making sulfur more accessible to the plants. Some autotrophic species also exhibit chemolithotrophic growth on inorganic sulfur compounds, such as *Acidothiobacillus, Thiosphaera*, and *Thiomicrospira*, but some heterotrophs, such as *Xanthobacter, Paracoccus, Pseudomonas*, and *Alcaligens* [80]. *Thiobacillus novellus* is considered an optional chemoautotroph, while chemoautotrophs are obligatory for *Thiobacillus ferrooxidans*, *T. thiooxidans*, *T. denitrificans* and *T. thioparus*. Inorganic sulfur compounds are reduced or partially oxidized by a heterogeneous group of sulfur bacteria. Thiobacilli plays a significant function in the sulfur oxidation and oxidation process, producing acidity, which aids to solubilize plant nutrients and enhances soil fertility [81]. *T. ferrooxidans* and *T. thioxidans* inoculation increased sulfur oxidation to pyrite and subsequently, rock phosphate solubilization [82]. Elemental sulfur and thiosulphate are oxidized by some fungi, which include a range of *Penicillium* species, *Epicoccum nigrum, Alternaria tenius, Scolecobasidium constrictum, Aspergillus, Aureobasidium pullulans*, and *Myrothecium cinctum* [83] (Table 2).

### 2.5. Zinc Solubilizers

Microorganisms provide micronutrients, such as copper, iron, and zinc by transforming the nutrients present in the soil into accessible fertilizers. The solubilization of zinc by microorganisms, viz., *T. thioxidans, Saccharomyces* sp., and *Bacillus subtilis. Bacillus* sp. can be used as a biofertilizer for zinc, can replace zinc sulfate, which is costly, and can be used in conjunction with compounds, such as zinc sulfide (ZnS), zinc oxide (ZnO), zinc carbonate (ZnCO_3_), and with cheap zinc compounds. Zinc, an important micronutrient for growth and metabolism, is needed by plants and microorganisms. Zinc is the main compound in an enzyme system, acting as a metal activator and co-factor for many enzymes [84]. *Gluconacetobacter diazotrophicus*, *Pseudomonas*, *Aspergillus niger*, and *Penicillium luteum* producing organic acid, such as gluconic acids; it derivatives as 2- and 2,5-keto-derivatives are Zn compound solubilizers [85]. *Thiobacillus ferroxidans, T. thioxidans*, and facultative oxidizers of thermophilic iron have enormous ability to solubilize sulfide ore zinc [86]. Bullen and Kemila [87] report that a few fungal sp. were affected by zinc. *Aspergillus niger* has been found to withstand a high level of zinc capable of growing below 1000 mg Zn and is used for zinc quantification in soils containing low zinc (Table 2).

## 3. Plant Growth Promoting Rhizobacteria (PGPR)

Soil inoculants or microbial inoculants are farm applications that stimulate the growth of plants and are beneficial microbes. Similar bacteria engage in a symbiotic association with crop plants, promoting both partners [14]. By stimulating growth regulators, these inoculants enhance plant nutrition and promote growth. Effective inoculants that increase the availability to plants of macronutrients, such as nitrogen and phosphorus, are nitrogen fixers and phosphate solubilizers [88]. These bacteria are classified as biofertilizers and rhizobacteria that promote plant growth. PGPR can be defined as free-living bacteria of the rhizosphere that enhance plant growth and function as specialists in biocontrol, biopesticides, or biofertilizers [10,89]. PGPR inoculants alternate with chemical fertilizers and pesticides as biofertilizers and/or antagonists of phytopathogens either directly or indirectly [90,91]. In generating various plant growth regulators and by mobilizing nutrients in soils, plant growth is stimulated. The PGPR action mechanisms are not fully known but are assumed to include: (i) Nitrogen fixation [92]; (ii) Organic phosphate and inorganic phosphate or other nutrient solubilization [93]; phytohormones, such as auxins, cytokinins [94], and gibberellins [95]; (iv) development of siderophores [96] and (v) plant defense by controlling or inhibiting phytopathogens, improving soil structure, and bioremediating contaminated soils by sequestering toxic heavy metals and destroying xenobiotic compounds (such as pesticides) [97,98]. The PGPR inoculant strains include species of *Azotobacter, Azospirillum, Agrobacterium, Acinetobacter, Alcaligenes, Arthrobacter, Acetobacter, Achromobacter, Aerobacter, Burkholderia, Beijerinckia, Bacillus, Clostridium, Delfitia, Erwinia, Enterobacter, Xanthomonas, Klebsiella, Flavobacterium, Micrococcus, Pantoea agglomerans, Paenibacillus macerans, Rhizobium, Pseudomonas, Rhodobacter, Serratia,* and *Rhodospirrilum* [23] (Figure 2 and Table 1).

### 3.1. Phytohormones

Within the control of plant growth and production, phytohormones, such as ethylene, gibberellins, auxins, abscisic acid (ABA), and cytokinins, play a key role [98]. Gutierrez-Manero et al. [99] have been reported that certain rhizospheric bacteria, such as *Bacillus licheniformis* and *Bacillus pumilus*, are capable of producing gibberellins. Various PGPR inoculants, such as *Azospirillum brasilense, Paenibacillus polymyxa, Arthrobacter giacomelloi, Bradyrhizobium japonicum, Bacillus licheniformi*, and *Pseudomonas fluorescens*, have been reported for the production of cytokinin [100,101]. Tissue expansion is encouraged by cytokinin, including cell division and cell enlargement in the plant. The root to shoot ratio is found to be reduced [102]. Auxin is an important phytohormone and controls multiple developmental processes, including root cell division, root initiation, and cell enlargement [103]. Indole-3-acetic acid (IAA) is produced by most rhizobacteria and stimulates plant growth promotion, especially root initiation and elongation [104]. IAA provided by PGPR is reported to increase root growth, modifying the plant (morphological functions) to uptake more nutrients from the soil (Table 3). Ethylene is another important phytohormone and plays a major role in the pathway of plant defense. Which inhibits root elongation and transport of auxins; abscission of different organs contributes to fruit maturation and promotes senescence [105]. *Azospirillum brasilense* produces ethylene, which probably facilitates the growth of root hair in tomato plants. Indeed *A. brasilense* inoculation had the mimicking effect of exogenous ethylene supply to plants, while this effect was inhibited by the addition of inhibitor for ethylene biosynthesis [106]. Lateral root extension and primary root elongation are promoted by gibberellins [107]. For the development of gibberellins, PGPR inoculants have been reported to produce gibberelline, several belonging to *Acinetobacter calcoaceticus, Achromobacter xylosoxidans, Azotobacter* sp., *Azospirillum* sp., *Rhizobia*, *Gluconobacter diazotrophicus*, *Bacillus* sp., and *Herbaspirillum seropedicae* [95]. Gutierrez-Manero et al. [99] documented that four different forms of GA are produced by *Bacillus licheniformis* and *B. pumilus* (Table 4).

### 3.2. Siderophore

Siderophore is an essential element for various biological processes in all organisms in the biosphere. Bacteria, fungi, actinomycetes, and certain algae developing under low iron stress synthesize siderophores. It is an iron-binding protein that has a molecular weight range of 400–1500 Da [108]. According to the functional group, they are divided into four families, i.e., carboxylates, catecholate, hydroxamates, and pyoverdines. About 270 siderophores were characterized structurally out of the 500 types [109]. Microbial siderophores help to identify the complex of bacterial ferric siderophores and enhance plant iron uptake [110] and are also significant in the presence of metals, such as nickel and cadmium, in the uptake of iron by plants [111]. Ferric ion absorption through siderophore is largely used in the soil, human body, and marine environments by pathogenic and nonpathogenic microorganisms.

Organisms producing siderophore includes bacteria (*Escherichia coli, Salmonella, Klebsiella pneumonia, Aerobacter aerogens, Mycobacterium* sp., *Yersinia*, *Enterobacter*, *Vibrio cholera*, *Aeromonas* and *Vibrio anguillarum*); Fungi include (*Trametes versicolor, Aspergillus versicolor, A. nidulans, Penicillium citrinum, P. chrysogenum, Ustilago sphaerogina, Rhizopus, Mucor, Rhodotorula minuta, Debaromyces* sp., and *Saccharomyces cerevisiae*) [44], Actinomycetes constitute (*Nocardia asteroids, Streptomyces griseus*, and *Actinomadura madurae*), and Algae (*Anabaena cylindrica* and *Anabaena flosaquae*) [45] (Table 2). Siderophore produced by Azospirillum inoculation; it can modify the root morphology by releasing substances that control plant growth [34,112].

### 3.3. Phytoremediation of Heavy Metals by PGPR

Phytoremediation is an energy proficient and cheap method of detoxification. Plant metabolism is influenced by reducing the metal bioavailability by absorbing them in the biomass of shoot [17]. Heavy metal phytoremediation is performed using PGPR. Agricultural activities and industrialization are the major reasons for metal contamination. Metal contamination of soil has a significant bearing on PGPR capacities. Upkeeping of metal homeostasis opposition in bacteria is achieved via the synthesis of binding proteins, sequestration, detoxification, reduced uptake, and active efflux [113]. Singh et al. [114] revealed that heavy metal contamination of soil caused the blocking of functional molecules, essential components dislodging in biomolecules, alteration of structure, and function of enzymes/protein. Heavy metals additionally repress biochemical processes, such as respiration and photosynthesis, resulting in a reduction of growth. The proliferation of root hair and drastic expansion of the surface area of root resulted after the inoculation of maize with *Azospirillum brasilense* [115]. Intense heavy metal tolerant *Pseudomonas putida*, and *P. fluorescens* PGPR have been successfully assessed under states of contaminated soils and hyperosmolarity [116]. In addition to PGPR, a significant part of phytoremediation is performed by mycorrhizal fungi [117]. *Streptomyces acidiscabies E13* strain applies positive growth developing effects in nickel contaminated soil of cowpea most likely by producing hydroxamate siderophores and binding of iron and nickel [112].

### 3.4. Antibiotic

Several bacterial antibiotics were used, such as aldehydes, hydrogen cyanide, alcohols, sulfides and ketones, diacetyl phloroglucinol, xanthobaccin, 2,4-diacetylphloroglucinol (DAPG), viscosinamide, mupirocin, pyocyanin, phenazine-1-carboxylic acid, phenazine-1-carboxamide (PCN), phenazine-1-carboxylic acid (PCA), hydroxy phenazines, zwittermicin A, butyrolactones, pyrrolnitrin, pyoluteorin, phenazine-1-carboxylic acid, kanosamine, oligomycin A, 2,4-diacetyl phloroglucinol, oomycin A, pyrrolnitrin [35,118], Agrocin 84, Agrocin 434 [119], herbicolin, phenazine [120], pyoluteorin, oomycin, siderophores, pyrrolnitrin, and hydrolytic enzymes, such as laminarinase, chitinase, Q-1,3-glucanase, lipase, and protease, as well as small molecules, such as hydrogen cyanide (HCN).

*Bacillus* sp. produces by circulin, polymyxin, and colistin, the majority of active compounds gram-negative and gram-positive bacteria along with many fungi [121]. Siddiqui et al. [122] reported the effect of Rhizobium to have higher colonization and siderophores production. *Pseudomonas* sp., producing HCN and DAPG, are contributing to the biological control of tomato canker bacteria [123]. Expression of various antibiotics by *Pseudomonas* was reported; phenazine, pyoluteorin [118], lipopeptide antibiotics [124] 2, 4-diacetylphloroglucinol [123] and bacterial antibiotic manufacturers are genetically manipulated, which is a powerful method for deciding their role in the suppression of diseases. *Arabidopsis thaliana* infected with *Pseudomonas syringae* gets protection against surfactin, which is produced by *Bacillus subtilis*. In addition, it protected the pathogen and also necessary for root colonization [125].

## 4. Biofertilizer Carrier

The carrier is the significant group of inoculants, which help deliver the appropriate volume of PGPM in superior physiological state. Assorted materials are used as inoculants carriers for having improved biological effectiveness, endurance, and shielding bacteria from abiotic and biotic stresses. The comprising elements of the carrier materials can be organic, inorganic, or synthetic. An appropriate carrier is chosen, depending on properties such as availability, low cost, easy use, packageability, and mixability. Additionally, the gas exchange must be allowed by the carrier, especially oxygen, which must have a high water-holding capacity and increased content of organic matter [126]. The physical form of biofertilizer is characterized by the carrier used. The mixture of soil carrier materials is utilized as dry inoculants, such as coal clays, peat, inert materials (bentonite, perlite, kaolin, silicates, and vermiculite), organic materials (sawdust, wheat bran, soybean meal, and composts), or inorganic soil (volcanic pumice or diatomite earth and lapillus). A variety of liquid inoculants, such as organic oils, oil-in-water suspensions, broth cultures, and minerals, can be utilized as carriers. Suitable carrier material for both bacterial inoculants and the plants themselves must be non-toxic. Moreover, Stephens and Rask [127]; Ferreira and Castro [128] expressed the properties of the carrier as promptly, plentifully, and locally assessable at less cost, easily sterilizable and neutral with a readily adjustable pH. The last choice of carrier incorporates properties, such as survival during storage, microbial multiplication, planting machinery, and sufficient cost, the general strategy of cultivation (Table 5).

Tilak [130] wrote about Farmyard manure (FYM) using blends, such as FYM + charcoal and soil, FYM + soil, and FYM + charcoal + soil, account for high viable counts of *Azospirillum* and survival up to 31 weeks. For the production of inoculants, carriers such as vermiculite clay, farmyard manure, coconut shell powder, teak leaf powder, and compost were used [131]. Locally accessible materials, such as coffee waste, soil, lignite, pressmud, and charcoal, were found to be superior to other carriers, which includes peat for Azospirillum, with the survival of 200 days and the decline rate in Azospirillum population was much lower in pressmud [132]. Singaravadivel and Anthoni Raj [133] reported that black ash plus husk mixture, pressmud, husk powder, black ash, and paddy husk were suitable and efficient carriers for Rhizobium and were also comparable with peat and lignite.

## 5. Biopesticides

Compared to conventional pesticides, biopesticides pose less risk to humans and the environment, gaining global attention as a new instrument for destroying or controlling pest species such as weeds, plant diseases, and insects [134,135]. Most biopesticides are advantageous for non-target biological safety and higher selectivity [136]. Biopesticides are types of pesticides that are produced from naturally occurring substances that control pests in an eco-friendly way via nontoxic mechanisms. Microorganism-derived biopesticides (*Nucleopolyhedrosis virus* and *Bacillus thuringiensis, Trichoderma*), plants (*Azadirachta* and *Chrysanthemum*), and animals (nematodes) contain their products (microbial products and phytochemicals) or by-products (semiochemicals) and live species (natural enemies) [137]. Biopesticides are categorized into three main categories: (i) pest-controlled microorganisms (microbial pesticides), (ii) naturally occurring pest-controlled substances (biochemical pesticides), and (iii) plant-controlled pesticides with added genetic material (PIPs). The use of biopesticides has increased by about 10% each year globally [138]. Biopesticides are natural or organically inferred agents, applied similarly to chemical pesticides, but accomplish environment-friendly pest management. All pest management products, particularly microbial agents, are helpful in control but need to be correctly formulated and used [139] (Figure 3 and Table 1 and Table 6).

### 5.1. Microbial Pesticides

Microbial pesticides are early developed and genetically modified. Organisms, such as algae, protozoans, fungi, viruses, or bacteria, are widely used. They develop pest-specific toxin, that causes disease, prevents the development of other microorganisms through antagonism or different nontoxic mechanism of action, compared to traditional chemical pesticides [147]. Normally, used microbial biopesticides are living microorganisms, pathogenic to the pest of interest, which include bioinsecticides (Bt), bioherbicides (Phytophthora), and bio fungicides (Pseudomonas, Trichoderma, and Bacillus) [148]. Microbial biopesticides comprise of microorganisms such as protozoa, bacteria, fungi, viruses, and oomycetes, which are generally used to control weeds, pestiferous insects, and plant pathogens biologically. In the market, 74% are guaranteed by bacterial biopesticides, 10% by fungal biopesticides, 10% by viral biopesticides, 8% by predator biopesticides, and 3% by others for a wide range of crops [149]. By generating toxic metabolites or various other modes, microbial pesticides can suppress different target pests [147]. The species used as microbial insecticides are generally nonpathogenic and nontoxic to all living organisms and not so firmly confined closely to the targeted pests [150].

#### 5.1.1. Bacteria

Bacterial biopesticides are used to monitor weeds, plant diseases, nematodes, and insects. Pest is controlled in various manners: delivering toxins, outcompeting and harming pathogens, promoting shoot and root growth, and producing anti-fungal compounds. Examples of bacterial biopesticides are *Pseudomonas syringae*, which controls bacterial spots, and *Bacillus thuringiensis* (Bt), which targets larvae. *Bacillus thuringiensis* (Berliner), the entomopathogenic bacterium, commonly recognized as a microbial biopesticide, which, during bacterial sporulation, generates crystal protein (d-endotoxin) when ingested by the susceptible insects triggers lysis of gut cells [140]. Spore formers, such as *Pseudomonas aeroginosa*, *Serratia marcesens*, *Bacillus thuringiensis*, and *Bacillus popilliae*, are used commercially for their efficacy and safety [141]. Pseudomonades, including *P. fluorescence*, *P. syringae*, and *P. aeruginosa*, are used to develop biopesticides. Some strains of *Pseudomonas aureofaciens* control plant pathogens, causing soft rots and damping-off [151]. Over half of mortality in *Helicoverpa armigera* and *Spodoptera litura* is by *Pseudomonas* sp., *Bacillus subtilis*, *B. megaterium*, and *B. amyloliquefaciens* [24]. Microbes like *B. subtilis, B. pumilus, B. licheniformis,* and *B. amyloliquefaciens* are marketed as biopesticides [142]. *Bacillus sphaericus* has been reported to have a dual role in larvicidal toxicity to *Culex pipien*, the blood-feeding mosquito, and the ability to excrete extracellular alkaline protease (AP) in the medium used for growth [152]. *Streptomyces griseoviridis* is the first biofungicides available to combat root infecting fungi in greenhouse crops. Despite such products’ long-term accomplishments, the global demand for new biopesticides remains [153,154]. *Bacillus thuringiensis* is sporulated, and it contains the proteins Cyt and Cry. Commercialized insecticides are products made up of 2% Bt, a combination of spores and protein crystals [155]. *Bacillus thuringiensis* can be more effective on *Aedes aegypti*, while the strain of *B. sphaericus* may be more effective on various mosquitoes, such as *Culex quinquefasciatus* [156]. In vegetables, it is recommended to use *Bacillus thuringiensis* (Bt) to manage insects, such as the velvet bean caterpillar, cabbage looper diamondback moth, and armyworm [143]. Sunitha et al. [157] found that the biopesticides based on *B. thuringiensis* are moderately active against *Metarhizium anisopliae*, while newer pesticides, such as spinosad and indoxacarb, were highly effective in controlling *Maruca vitrata*. Schunemann et al. [143] recommended various trade products of *B. thuringiensis* to control insect pests of agriculture, including mosquito species. Most formulations of spore-crystal toxins are obtained from a variety of strains, such as *B. thuringiensis var. kurstaki, B. thuringiensis var. tenebrionis, B. thuringiensis var. israelensis, B. thuringiensis var. aizawai,* and *B. thuringiensis var. San Diego* [143].

#### 5.1.2. Fungi

In killing mites, weeds, nematodes, insects, or other fungi, new fungal biopesticides are used. Like bacteria, they produce toxins, such as bacteria, that outcompete targeted pathogens. These can also paralyze plant pathogens or insects by attacking them. *Trichoderma harzianum*, targeting Pythium, Rhizoctonia, and Fusarium, is also a fungicide [158]. Fungal species, such as *Paecilomyces fumosoroseus, Beauveria bassiana, Verticillium lecani, Nomuraea rileyi,* and *Metarhizium anisopliae* are used in insect control [25]. Beauverin peptide isolated from *Beauveria bassiana* is active against larvae of mosquito [159]. Fungal pathogens *Metarhizium anisopliae* and *Beauveria bassiana* have a lengthy- history in the perspective of agricultural pests. Current molecular techniques allow for the characterization and monitoring isolates of fungi, as well as for recognizing fungal isolates in the environment [160,161]. The codling moth and colorado potato beetle were regulated using *Beauveria bassiana* [162]. Biopesticides, such as *M. anisopliae* are commercially available, which controls several insect species [163]. Destruxins, a toxin produced by *M. anisopliae*, which has two separate virulence mechanisms, includes invading and destroying the insects, and third mechanisms by invading the ticks by a strategy of integument breakdown [164].

#### 5.1.3. Nematodes

Several round colorless parasites, nematodes, and microscopic worms of the plant cause severe crop damage. Though targeting plants, some are essentially advantageous in attacking soil-dwelling insect pests, such as root weevils and cutworms [155]. Nematode biopesticides, such as *Steinernema* sp. and *Heterorhabditis* sp., that attack the hosts as contagious juveniles (IJs) are widely used [165]. *Heterorhabditis megidis*, *H. bacteriophora, Steinernema scapterisci, S. carpocapsae, S. riobrave, S. glaseri*, and *S. feltiae* are common entomogenous nematodes used as insecticides [144].

#### 5.1.4. Protozoa

Protozoans are single-celled organisms surviving both in soil and water. Most species are parasites of insects, typically feeding on bacteria, while others feed on organic decay. More than any other insects, lepidopteran and orthopteran, hoppers especially are killed by Vairimorpha and Nosema comparing to other insects [166]. Nosema locustae spores enter and feed on the grasshopper body cavity. Mortality can take up to 3–4 weeks [167].

#### 5.1.5. Viruses

Baculoviruses are a family of viral biopesticides believed to infect insects and arthropods related to them. Potential pesticides are the family Baculoviridae. This biopesticide is used in many parts of the world for the prevention of destructive caterpillar pests [168]. Nucleopolyhedro virus (NPVs) and Granulovirus (GVs) are found to be the two main genera of the Baculoviridae family [169]. These viruses are valuable, causing minimum damage, suitable for the crop, and management of pests, since only a few species of Lepidoptera larvae are infected, due to host specificity. The corn earworm *Heliothis*/*Helicoverpa* sp. by nuclear polyhedrosis virus and the codling moth of *Cydia pomonella* by granulosis virus are some examples. In contrast with traditional synthetic insecticides, Baculoviruses can control lepidopteran pests causing slight or no damage to the targeted species. The first viral biopesticide detected is the Heliothis nuclear polyhedrosis virus (NPV) [145]. Expression vectors, developed based on baculoviruses, were used in the production of viral pesticides using *Autographa californica nucleopolyhedro virus (AcMNPV)*. *Autographa gemmatalis* control the soybean velvet bean caterpillar [146].

### 5.2. Biochemical Pesticides

Biochemical pesticides are equivalent to the naturally occurring or compounds, derived synthetically, that are used in pest control. The influence of growth and develop-ment of insect pests is achieved by the biochemical pesticides which are nontoxic in action destroying or attacking pest. Pheromones are substances that attract or repel pest or growth regulators of plant growth produced by biochemical pesticides that interfere in mating and growth of pests, including elements, such as insect sex pheromones interfering in mating, as well as attracting insect pests to traps using extracts of the scented plant. Chemical substances, such as pheromones, are emitted by living organisms that are used in sending messages to the same species individuals of mostly opposite sex [170]. Minimal crop damage can be achieved by using sex pheromones and plant protection measures by recognizing the crops and insects for further required action. The remarkably effective synthetic attractant is used in a low population, often use pheromone traps or a technique called “attracting and killing”.

### 5.3. Plant Incorporated Protectants (PIPS)

Substances producing pesticides (PIPs) are introduced into the target crop plant ge-nome, thereby providing the plant with capability of killing the pest. Scientists insert a insecticidal protein gene of *Bacillus thuringiensis* insecticidal protein into the plant’s genetic material thereby allow the plants to kill the pest. Environmental Protection Agency controls the protein, genetic material, and not the plant itself [170].

## 6. Conclusions

Biofertilizers based on microbial inoculants are attractive because they act in fixing nitrogen, phosphate, sulfate, potassium, zinc, and solubilize nutrients and enhance plant growth by hormonal action or antibiosis and decomposing organic residues. Plant reinforcers and phytostimulators can be used by plants to improve their growth when insufficient quantities of nitrogens are present. Moreover, they emerged from the soil and appeared to be competent in the rhizosphere. Plant growth-promoting rhizobacteria with numerous activities, such as nitrogen fixation, phytohormone production, micro- and macro-mineral solubilization, enzymes production, or fungicidal compounds of antibiotics synthesis. Siderophores, a competition with detrimental microorganisms, have bioremediation potentials by detoxifying contaminants, such as pesticides, heavy metals, and regulate phytopathogens, as biopesticides. They also improve and maintain the soil rhizosphere biologically by microbes, such as bacteria, fungi, algae, and actinomycetes. This review discusses the multiple activities of single organisms or consortia, such as nitrogen-fixing, phosphate, sulfate, and zinc solubilization, through enzyme and acid production. The effect of microorganisms as biofertilizers and the role of biopesticides enhance plant growth by rendering them as tolerant to pests and to improve the crop health and food safety.

## Figures and Tables

**Figure 1 biology-10-01111-f001:**
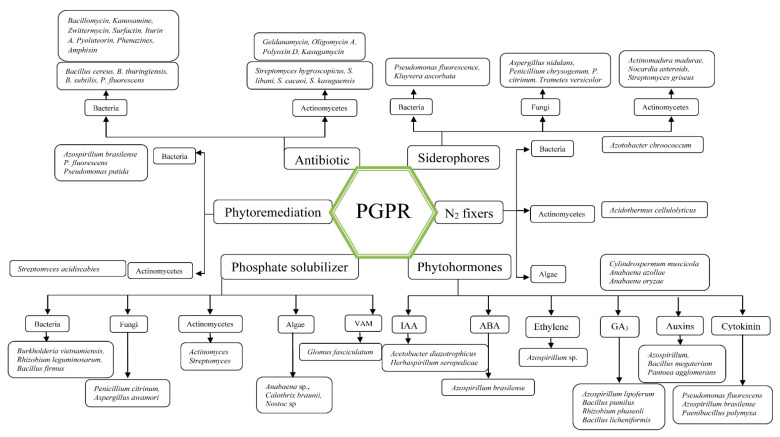
PGPR classification and involved microorganisms.

**Figure 2 biology-10-01111-f002:**
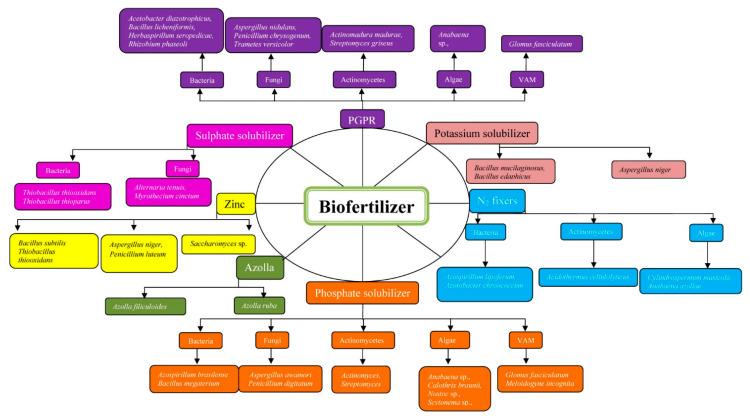
Biofertilizer classification and involved microorganisms.

**Figure 3 biology-10-01111-f003:**
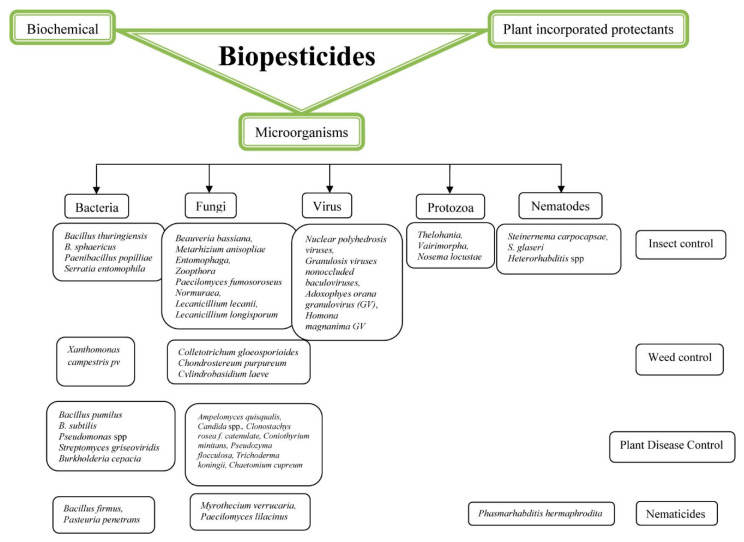
Biopesticides classification and involving microorganisms.

**Table 1 biology-10-01111-t001:** Microbial inoculants used as biofertilizers, Plant Growth Promoting Rhizobacteria, and Biopesticides.

Biofertilizer	PGPR	Biopesticide	References
*Rhizobium, Azotobacter, Azospirillum brasilense, Azospirillum lipoferum, Azotobacter chroococcum, Acetobacter diazotrophicus, Bacillus licheniformis, B. megaterium, B. mucilagenosus, B. edaphicus, B. subtilis, Actinomyces, Streptomyces, Herbaspirillum seropedicae, Rhizobium phaseoli, Thiobacillus thioxidans, Glomus fasciculatum, Blue Green Algae (BGA)*, and *Azolla.*	*Acetobacter, Aeromonas hydrophila, Azotobacter, Achromobacter, Alcaligenes, Anabaena, Arthrobacter, Azoarcus, Azospirillum brasilense, A. irakense, A. lipoferum, Azotobacter, Acinetobacter calcoaceticus, A. baumannii, Bacillus polymyxa, Beijerinckia, Burkholderia gladioli, Burkholderia cepacia, Clostridium, Derxia, Enterobacter, Erwinia* spp., *Ewingella americana, Escherichia vulneris, Flavobacterium, Frankia, Gluconacetobacter, Klebsiella, Mycobacterium phlei, Proteus penneri, Pseudomonas fluorescens, P. luteola, P. alcaligenes, P. putida, Rhizobium leguminosarum, Rahnella aquatilis, Serratia plymuthica, S. ficaria, Sinorhizobium, Shigella* spp., *Vibrio fluvialis*, and *Zoogloea*	*Bacillus thuringiensis, B. thuringiensis var. kurstaki (Bt), B. thuringiensis var. israelensis (Bt), B. thuringiensis var. tenebrionis, B. thuringiensis var. aizawai, B. thuringiensis japonensis, B. popilliae, B. lentimorbus, B. sphaericus, B. pumilus, B. subtilis, B. firmus, Burkholderia cepacia, B. amyloliquefaciens, B. licheniformis, Erwinia amylovora, Pasteuria penetrans, Pasteuria usage, Pseudomonas* spp., *Streptomyces griseoviridis*, and *Xanthomonas campestris pv. poannua,*	[22,23,24,25]

**Table 2 biology-10-01111-t002:** Microbial inoculants in phosphate, sulphate, zinc solubilizer and nitrate, siderophore producers are used as biofertilizer.

Types	Bacteria	Fungi/VAM	Actinomycetes	Cyanobacteria/Yeast	References
PSM *	*Alcaligenes* sp., *Aerobacter aerogenes*, *Achromobacter* sp., *Actinomadura oligospora*, *Agrobacterium* sp., *Azospirillum brasilense*, *Bacillus circulans*, *B*.*cereus*, *B.fusiformis*, *B. pumilus*, *B. megaterium*, *B. mycoides*, *B. polymyxa*, *B. coagulans*, *B.chitinolyticus*, *B. subtilis*, *Bradyrhizobium* sp., *Brevibacterium* sp., *Citrobacter* sp., *Pseudomonas putida*, *P. striata*, *P. fluorescens*, *P. calcis*, *P. corrugate*, *Flavobacterium* sp., *Nitrosomonas* sp., *Erwinia* sp., *Micrococcus* sp., *Escherichia intermedia*, *Enterobacter asburiae*, *Serratia phosphoticum*, *Nitrobacter* sp., *Thiobacillus ferroxidans*, *T*. *thioxidans*, *Rhizobium meliloti,* and *Xanthomonas* sp.	*Aspergillus awamori, A. niger, A. terreus, A. flavus, A. nidulans, A. foetidus, A. wentii, Fusarium oxysporum, Alternaria teneius, Achrothcium* sp., *Penicillium digitatum, P. lilacinium, P. balaji, P. funicolosum, Cephalosporium* sp., *Cladosprium* sp., *Curvularia lunata*, *Cunnighamella*, *Candida* sp., *Chaetomium globosum*, *Humicolainslens*, *H. lanuginosa*, *Helminthosporium* sp., *Paecilomycesfusisporous*, *Pythium* sp., *Phoma* sp., *Populosporamytilina*, *Myrotheciumroridum*, *Morteirella* sp., *Micromonospora* sp., *Oideodendron* sp., *Rhizoctonia solani*, *Rhizopus* sp., *Mucor* sp., *Trichoderma viridae*, *Torula thermophila*, *Schwanniomyces occidentalis,* and *Sclerotium rolfsii*. *Glomus fasciculatum (VAM)*	*Actinomyces* sp. and *Streptomyces* sp.	*Anabaena* sp., *Calothrix braunii*, *Nostoc* sp., and *Scytonema* sp.,	[22,34,35,37,38,39,40,41,42,43,44,45]
SSM *	*Acidothiobacillus*, *Thiomicrospira*, *Thiosphaera*, *Paracoccus*, *Xanthobacter*, *Alcaligenes, Pseudomonas*, *Thiobacillus thiooxidans*, *T. ferrooxidans*, *T. thioparus, T. denitrificans,* and *T. novellus*	*Aureobasidium*, *Epicoccum*, *Penicillium*, *Aspergillus*, *Alternariatenuis*, *Aureobasidiumpullulans*, *Epicoccumnigrum*, *Scolecobasidiumconstrictum,* and *Myrotheciumcinctum*			
NO_3_ *	*Azospirillum lipoferum, A. brasilense, Azoarcus, Azotobacter chroococcum, A. peroxydans, A. nitrogenifigens, Rhizobium, Bradyrhizobium, Sinorhizobium, Azorhizobium, Mesorhizobium, H. seropedicae, H. rubrisubalbicans Burkholderia* sp., *Rhizobium leguminosarum bv. trifolii, B. vietnamiensis, Gluconacetobacterkombuchae, G. johannae, G. azotocaptans, G. diazotrophicus,* and *Swaminathania salitolerans*		*Acidothermus cellulolyticus*	*Cylindrospermum musicola* and *Anabaena azollae*	
Siderophore	*Bacillus* sp., *Ochrobactrum, Kluyvera ascorbata, Salmonella, Enterobacter, Yersinia, Mycobacterium, B. megaterium, Ochrobactrum anthropi, Proteus vulgaris, Pseudomonas fluorescence, P. putida, Escherichia coli, Salmonella, Klebsiella pneumoniae, Vibrio cholerae, V. anguillarum, Aeromonas, Aerobacter aerogenes, Yersinia,* and *Mycobacterium*	*Aspergillus nidulans, A. versicolor, Penicillium chrysogenum, P. citrinum, Mucor, Rhizopus, Trametes versicolor, Ustilago sphaerogina, Debaromyces* sp.*,* and *Rhodotorula minuta*	*Nocardia asteroids, Streptomyces griseus,* and *Actinomadura madurae*	*Saccharomyces cerevisiae* (Yeast)	
ZSB *	*Bacillus subtilis, Gluconacetobacter diazotrophicus, Thiobacillus thioxidans, and T. ferroxidans*	*Aspergillus niger* and *Penicillium luteum*		*Saccharomyces* sp. (Yeast)	

* PMS—Phosphate Solubilizing Microorganism, SSM—Sulphte Solubilizing Microorganism, ZSB—Zinc Solubilizing Microorganism.

**Table 3 biology-10-01111-t003:** Plant growth promoting substance (acids) producing microorganisms.

Microorganisms	Acids	References
*Bacillus pumils, B. subtilis, B. licheniformis, B. megaterium BHUPSB14,* and *Paenibacillus polymyxa*	Gibberellins, Ethylene, Cytokinin, and ACC deaminase	[44,45,100]
*Pseudomonas tabaci, P. putida, P. syringae, P. fluorescens, P. fluorescens* G20-18, *P. fluorescens* BHUPSB06, *P. aeruginosa, P. cepacia,* and *P. corrugata*	Ethylene, Indole-3-acetic acid, Cytokinin, and ACC deaminase	
*Rhizobium leguminosarum*	Indole-3-acetic acid, Cytokinin, and HCN	
*Azospirillum brasilense* and *A. lipoferum,*	Indole-3-acetic acid, Zeatin, and ethylene, Gibberellic acid (GA3), and Abscisic acid (ABA)	
*Rhizobacterial isolates*	Auxins	
*Aeromonas veronii, Agrobacterium* sp., *Bradyrhizobium* sp., *Comamonas acidovorans, Azotobacter chroococcum, Mesorhizobium ciceri*, *Azospirillum amazonense, Rhizobium* sp., *Azotobacter* sp., *Kebsiellaoxytoca, Erwinia herbicola, Bacillus subtilis, Serratia marcescens,* and *Enterobacter asburiae*	Indole-3-acetic acid	
*Alcaligenes piechaudii* and *Enterobacter cloacae*	Indole-3-acetic acid, ACC deaminase	
*Variovorax paradoxus*	ACC deaminase	
*Pantoea agglomerans* and *Pantoea herbicola*	IAA and Auxin	
*Gluconobacter diazotrophicus*	GA3, indole-3-acetic acid, and gibberellin GA1	

**Table 4 biology-10-01111-t004:** Plant growth promoting and bio controlling enzymes and acids producing phosphate solubilizing microbes.

Microorganisms	Enzymes	Acids	References
*Bacillus circulans, B.cereus, B. fusiformis, B.pumilus var.2, B. megaterium, B. mycoides, B. polymyxa, B. coagulans B. chitinolyticus, B. subtilis, B. subtilisvar.2, B. licheniformis, B. amyloliquefaciens, B. atrophaeus, Paenibacillus macerans,* and *B. japonicum*	Phytase and D-a-glycerophosphate	Lactic, malic, citric, itaconic, isovaleric, isobutyric, acetic, gluconic, propionic, heptonic, Caproic, Isocaproic, Formic, valeric, succinic, Oxalic, oxalacetic, malonic, and IAA	[95,100,101]
*Bradyrhizobium* sp.,	Phytate	IAA	
*Burkholderia cepacia, Citrobacter* sp., and *Citrobacter freundii*	Acid phosphatase	Gluconic acid	
*Escherichia intermedia* and *E. freundii*	-	Lactic	
*Enterobacter asburiae, E. aerogenes, E. cloacae, E. aerogenes,* and *E. intermedium*	Acid phosphatase	Lactic, itaconic, isovaleric, isobutyric, acetic, 2-ketogluconic, gluconic, succinic, acetic, glutamic, oxaloacetic, pyruvic, malic, fumaric, and alpha-ketoglutaric	
*Pseudomonas putida, P. striata, P. fluorescens, P. calcis, P. mendocina,* and *P. aeruginosa*	Acid phosphatase, Phytase, and Phosphonoacetate hydrolase	Lactic, malic, citric, gluconic, 2-ketogluconic acid, and tartaric	
*Proteus mirabilis*	Acid phosphatase		
*Serratia phosphoticum* and *S. marcescens*	Acid phosphatase	Gluconic acid and IAA	
*Rhizobium meliloti, R. leguminosarum, R. leguminosarum bv.phaseoli, R. leguminosarum bv. Trifolii,* and *R. leguminosarum bv. Viciae*	Phytate	2-ketogluconic acid, HCN, and IAA	
*Klebsiella aerogenes*	C-P Lyase		
*Sinorhizobium meliloti*	Phytate	IAA, malic, succinic, and fumaric	
*Stenotrophomonas maltophilia*		Gluconic acid	
*Mesorhizobium cireri* and *M. mediterraneum*	Phytate		
*Acetobacter* sp.		Gluconic acid	

**Table 5 biology-10-01111-t005:** Microbial inoculants carrier types as biofertilizer.

Materials	Category	Reference
Preservative and Culture media ( liquid and powder)	Bacterial cultures (lyophilized)	Bashan and de-Bashan, [129]
Alginate and xanthan gum	Biopolymer	
Black ash, paddy husk, black ash plus husk mixture, husk powder and pressmud, soybean and peanut oils, farmyard manure, plant debris, wheat bran, composts, spent mushroom composts, sugar industry waste, agricultural waste material, soybean meal, coconut shell powder, and teak leaf powder	Waste materials (Plant)	
Lignite, pressmud, charcoal, inorganic soil, coal, clays and peat	Soils	
Carrageenan, polyacrylamide, calcium sulfate, polysaccharide-like alginate, ground rock phosphate, vermiculite, and perlite	Inert materials	

**Table 6 biology-10-01111-t006:** Microbial-based biopesticides.

Micro Organisms	Pest Control	Weed Control	Plant Disease Control	Nematicides Control	Fungicides	Reference
Bacteria	*Bacillus thuringiensis, B. thuringiensis var. kurstaki, B. thuringiensis var. israelensis, B. thuringiensis var. tenebrionis, B. thuringiensis var. aizawai, B. thuringiensis japonensis, B. popilliae, B. lentimorbus, B. sphaericus, Erwinia amylovora,* and *B. pumilus*	*Xanthomonas campestris pv. Poannua*	*Bacillus pumilus, B. subtilis, Pseudomonas* spp.*, Streptomyces griseoviridis,* and *Burkholderia cepacia*	*Bacillus firmus, Pasteuria penetrans,* and *Pasteuria usage*	*Bacillus amyloliquefaciens, B. licheniformis, B. pumilus,* and *B. subtilis*	[24,25,137,140,141,142,143,144,145,146].
Fungi	*Beauveria bassiana, Metarhizium anisopliae, Entomophaga, Zoopthora, Paecilomyces fumosoroseus, Normuraea, Lecanicillium lecanii, L. longisporum, Lagenidium giganteum,* and *Verticillium lecanii*	*Colletotrichum gloeosporioides, Chondrostereum purpureum* and *Cylindrobasidium laeve*	*Ampelomyces quisqualis, Candida* sp.*, Clonostachys rosea f. catenulate, Coniothyrium minitans, Pseudozyma flocculosa, Trichoderma harzianum, T. koningii, T. viride,* and *Chaetomium cupreum*	*Paecilomyces lilacinus, Myrothecium verrucaria, Verticillium chlamydosporium,* and *Pochonia chlamydosporia*	
Protozoa	*Nosema locustae, Thelohania,* and *Vairimorpha*				
Nematodes	*Steinernema feltiae, S. carpocapsae, S. glaseri, S. riobravis,* and *Heterorhabditis heliothidis*				
Virus	*Tussock moth* NPV*, Pine sawfly* NPV*, Granulosis viruses, Codling moth granulosis virus *(GV)*, Gypsy moth nuclear polyhedrosis (NPV), Nuclear polyhedrosis viruses, non-occluded baculoviruses, Adoxophyes orana granulovirus *(GV)*+ Homona magnanima* GV*, Cydia pomonella granulovirus, Nucleopolyhedrovirus Neodiprion abietis, Heliothis zea* NPV*, Anagrapha falcifera* NPV*, Spodoptera exigua* NPV*, Mamestra configurata* NPV*, Ectropis obliqua hypulina* NPV*, Laphygma exigua NPV, Prodenia litura* NPV*, Buzura suppressaria* NPV*, Gynaephora ruoergensis* NPV*, Mythimna separata* NPV*, Periplaneta fuliginosa densovirus virus, Pieris rapae* GV*, Mythimna separata* GV, and *Plutella xylostella* GV

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
