# Peer review of "Utilization of Microbial Consortia as Biofertilizers and Biopesticides for the Production of Feasible Agricultural Product"

_biology, 2021, doi:10.3390/biology10111111_

Round 1

Reviewer 1 Report

Microbes are ubiquitous in the environment and play a major role in the ecosystem and are even excellent for plant growth promotion for agro-environmental sustainability. Agricultural sustainability is facing to be a formidable task by using chemical-based fertilizers and pesticides in order to increase the yield of the crop plants. 
The soil microbiome has found diverse and complex habitats, which consist of billions of bacteria, fungi, and other living organisms. Beneficial microbes play an essential role in nutrient cycling and plant shielding from destructive effects of biotic and abiotic stresses. Intensive farming practices lead to an increase in crop production, but they also have detrimental effects on the biological and physiological properties of soils. The macronutrients for plant growth are generally provided via chemical fertilizers. In comparison to chemical and synthetic fertilizers, biofertilizers and biopesticides improve plant growth and crop productivity in an eco-friendly way. Along with plant growth promotion, beneficial microbes could be used for mitigation of diverse abiotic stresses using diverse plant growth promoting mechanisms. Thus, using beneficial microbiomes for sustainable agriculture is gaining vast attention worldwide.
In the present work, the authors presented the possibility of using utilizing microbial consortia as biofertilizers and biopesticides for the production of feasible agricultural product. This research is interesting and clearly provides new data valuable for the research community.

GENERAL COMMENTS:
The paper title is well stated, it is informative and concise.
ABSTRACT, INTRODUCTION, AND CONCLUSIONS
Abstract and introduction is well written.

In general, the DISCUSSION is correct.
The items of literature included in the paper are adequate to the subject of the paper. The number of publications cited is too long. Please refer to the most important data.

The text of the manusctipt is not formatted correctly. Incorrectly formatted text in tables makes them unreadable. Please pay attention to the quality of the presented graphs, e.g. the graphs have the wrong resolution, are unreadable and cannot be published in this form.

Author Response

Response to Anonymous Review report (Interactive comment on Journal of Biology for review entitled “A Review on utilizing microbial consortia as biofertilizers and biopesticides for the production of feasible agricultural product).

We would like to thank the reviewer for the comments, since His/her recommendations contribute to clarify specific aspects of the review. His /her considerations have been taken into account for the revised version of paper. We provide below the answers to the specific comments made.

Reviewer 1:

Q1. GENERAL COMMENTS:

The paper title is well stated, it is informative and concise.

ABSTRACT, INTRODUCTION, AND CONCLUSIONS

Abstract and introduction is well written.

In general, the DISCUSSION is correct.

Reply: Thanks for the positive comments about worth of its content. This gives enthusiastic approach for our advance paper work.

Q 2. The items of literature included in the paper are adequate to the subject of the paper. The number of publications cited is too long. Please refer to the most important data.

Reply: We accept the suggestion and some of the references were deleted for getting the revised manuscript with a concise manner.

Q 3. The text of the manuscript is not formatted correctly. Incorrectly formatted text in tables makes them unreadable. Please pay attention to the quality of the presented graphs, e.g. the graphs have the wrong resolution, are unreadable and cannot be published in this form.

Reply: As per the reviewer suggestion, necessary corrections have been carried out in the revised manuscript.

Reviewer 2 Report

Please find the comments and suggestions on the manuscript entitled "A Review on utilizing microbial consortia as biofertilizers and biopesticides for the production of feasible agricultural product".

  1. Abstract – it does not reflect the problem addressed by the authors. Objectives were not clear and the drafting is poorly constructed—it's hard to correlate. Please rewrite it and also take help of some native English speaker for better presentation—
  2. Introduction- Not understandable and the correlation between the paragraphs is missing--, the flow content is also not found. The English are also an issue in the whole manuscript--- rewrite the whole introduction section focusing on the subject addressed--
  3. Figure 1- the quality is not clear- needs improvements ---
  4. Tables were without citations which do not reflect in scientific sense please do the needful and change accordingly ---
  5. Overall after going through the review article I suggest authors please rework on the whole review article, especially working on English language and content as of now there are no correlations seen in the manuscript— Look forward to a revised improved version for suggestions and improvements -- 

Author Response

Response to Anonymous Review report (Interactive comment on Journal of Biology for review entitled “A Review on utilizing microbial consortia as biofertilizers and biopesticides for the production of feasible agricultural product).

We would like to thank the reviewer for the comments, since his/her recommendations contribute to clarify specific aspects of the review. His /her considerations have been taken into account for the revised version of paper. We provide below the answers to the specific comments made.

Reviewer 2:

Q 1.     Abstract – it does not reflect the problem addressed by the authors. Objectives were not clear and the drafting is poorly constructed—it's hard to correlate. Please rewrite it and also take help of some native English speaker for better presentation—

Reply: As per reviewer suggestion and recommendation we have rewritten the abstract, objectives with the help of native English speaker.

 Q 2. Introduction- Not understandable and the correlation between the paragraphs is missing--, the flow content is also not found. The English are also an issue in the whole manuscript--- rewrite the whole introduction section focusing on the subject addressed—

Reply: Necessary corrections have been incorporated in the revised manuscript.

Q 3. Figure 1- the quality is not clear- needs improvements ---

Reply: We had carried out the correction as per the reviewer comments.

Q 4. Tables were without citations which do not reflect in scientific sense please do the needful and change accordingly ---

Reply: We had added references of the tables as per the reviewer comments.

Q 5. Overall after going through the review article I suggest authors please rework on the whole review article, especially working on English language and content as of now there are no correlations seen in the manuscript— Look forward to a revised improved version for suggestions and improvements –

Reply: Necessary corrections for English language and content editing work have been performed the revised manuscript.

Round 2

Reviewer 2 Report

Authors have addressed the suggestions provided on the manuscript entitled "A Review on utilizing microbial consortia as biofertilizers and biopesticides for the production of feasible agricultural product". However, I have still some minor reservations.

  • Line no-580- a mixture of microbes- it does not sound well instead of using it please use either co-inoculation or consortium is better ---
  • I, suggest please replace the mixture of microbes everywhere in the manuscript with consortium or Co-inoculation of microbes –
  • If there have been a separate section dealing with Co-inoculated microbes that would have been better to understand – just like authors have done for single microbes with tabulation format---

Thanks for the revised version

Author Response

Response to Anonymous Review report (Interactive comment on Journal of Biology for review entitled “Utilization of microbial consortia as biofertilizers and biopesticides for the production of feasible agricultural product).

We would like to thank the reviewer for the comments, since his/her recommendations contribute to clarify specific aspects of the review. His /her considerations have been taken into account for the revised version of paper. We provide below the answers to the specific comments made.

Reviewer 2:

Authors have addressed the suggestions provided on the manuscript entitled "A Review on utilizing microbial consortia as biofertilizers and biopesticides for the production of feasible agricultural product". However, I have still some minor reservations.

Q 1. Line no-580- a mixture of microbes- it does not sound well instead of using it please use either co-inoculation or consortium is better ---

I suggest please replace the mixture of microbes everywhere in the manuscript with consortium or Co-inoculation of microbes –

Reply: Yes, we had changed the whole manuscript on the appropriate word “a mixture of microbes” to the consortium as per the reviewer comments.

Q 2. If there have been a separate section dealing with Co-inoculated microbes that would have been better to understand – just like authors have done for single microbes with tabulation format---

Reply: Since we provide the separate table for Co-inoculated microbes in the manuscript, it seems to be felt that repetitions of microbial names. Hence for better avoiding confusion separate section cannot be inserted in the revised manuscript.